# Isotopic Tracers Unveil Distinct Fates for Nitrogen Sources during Wine Fermentation with Two Non-*Saccharomyces* Strains

**DOI:** 10.3390/microorganisms8060904

**Published:** 2020-06-16

**Authors:** Ying Su, Pauline Seguinot, Audrey Bloem, Anne Ortiz-Julien, José María Heras, José Manuel Guillamón, Carole Camarasa

**Affiliations:** 1Departamento de Biotecnología de Alimentos, Instituto de Agroquímica y Tecnología de Alimentos (IATA), Consejo Superior de Investigaciones Científicas (CSIC), 46980 Paterna-Valencia, Spain; ying.su@iata.csic.es (Y.S.); guillamon@iata.csic.es (J.M.G.); 2UMR SPO, Université Montpellier, INRAE, Montpellier SupAgro, 34060 Montpellier, France; seguinot.pauline@gmail.com (P.S.); audrey.bloem@umontpellier.fr (A.B.); 3Lallemand SAS, 31700 Blagnac, France; ajulien@lallemand.com; 4Lallemand Bio, 28521 Madrid, Spain; jmheras@lallemand.com

**Keywords:** ^15^N- and ^13^C-isotope labelling, quantitative analysis of metabolism, nitrogen resource management, *Torulaspora delbrueckii*, *Metschnikowia pulcherrima*

## Abstract

Non-*Saccharomyces* yeast strains have become increasingly prevalent in the food industry, particularly in winemaking, because of their properties of interest both in biological control and in complexifying flavour profiles in end-products. However, unleashing the full potential of these species would require solid knowledge of their physiology and metabolism, which is, however, very limited to date. In this study, a quantitative analysis using ^15^N-labelled NH_4_Cl, arginine, and glutamine, and ^13^C-labelled leucine and valine revealed the specificities of the nitrogen metabolism pattern of two non-*Saccharomyces* species, *Torulaspora delbrueckii* and *Metschnikowia pulcherrima.* In *T. delbrueckii*, consumed nitrogen sources were mainly directed towards the *de novo* synthesis of proteinogenic amino acids, at the expense of volatile compounds production. This redistribution pattern was in line with the high biomass-producer phenotype of this species. Conversely, in *M. pulcherrim*a, which displayed weaker growth capacities, a larger proportion of consumed amino acids was catabolised for the production of higher alcohols through the Ehrlich pathway. Overall, this comprehensive overview of nitrogen redistribution in *T. delbrueckii* and *M. pulcherrima* provides valuable information for a better management of co- or sequential fermentation combining these species with *Saccharomyces cerevisiae*.

## 1. Introduction

Traditionally, wine fermentation is a process in which sugars are converted into ethanol through interactions between microorganisms, mainly yeast species, also imparting pleasant notes to wines [1]. However, some spoilage microorganisms may cause problematical wine fermentations. In 1883, Emil Christian Hansen successfully isolated the first pure yeast culture [2], which started a new era for yeast research and utilisation. In modern oenology, inoculation of pure cultures of *Saccharomyces cerevisiae* is widely used as an efficient way to prevent the growth of spoilage non-*Saccharomyces* species, thus ensuring the completion of fermentation and a stable wine quality [3,4,5]. Nevertheless, in the last decade, more attention has been paid to non-*Saccharomyces* yeasts for their technological properties of interest in winemaking. Two of the most studied and commercialised of these species are *Torulaspora delbrueckii* and *Metschnikowia pulcherrima* [6], used to reduce the production of ethanol and volatile acids while, on the contrary, increasing glycerol concentration and wine aromatic complexity (for a review see [7], and references therein). However, due to their low tolerance to ethanol or SO_2_, non-*Saccharomyces* yeasts are unable to consume all the sugars available in grape juices, and thus are often used together with *S. cerevisiae* in co- or sequential fermentations to ensure complete wine fermentation. The main consequence of these combined inoculation procedures is that competition for nutrients, mainly nitrogen resources, may occur between the two species involved, resulting in fermentation issues. Furthermore, it has been widely reported that the quantity and quality of nitrogen sources in grape must directly affect the formation of volatile compounds and determine the wine organoleptic profile [8]. Therefore, understanding nitrogen metabolism during fermentation is of utmost significance.

In *S. cerevisiae,* many studies have been carried out to draw a comprehensive picture of both the mechanisms responsible for the import and use of nitrogen sources during winemaking, and their regulation. Nitrogen is imported into cells by different transporters, controlled by three regulation systems: the nitrogen catabolic repression (NCR), the SPS sensors, and the general amino acids control (GAAC) [9,10,11]. The combination of these different regulation mechanisms explains the order in which nitrogen sources are consumed in *S. cerevisiae* [12]. Once inside the cells, amino acids can be either directly incorporated into proteinogenic amino acids or degraded. A recent study using labelled amino acids showed that the amount of amino acids directly incorporated into biomass is low, except for arginine, histidine, and lysine [13]. The largest part of amino acids is catabolised to release ketoacids and nitrogen as ammonium or glutamate. Ketoacids can then be used for the *de novo* synthesis of amino acids, catabolised towards the central carbon metabolism (CCM) or towards the production of aroma compounds through the Ehrlich pathway. Ammonium and glutamate are redirected towards the core nitrogen metabolism composed of reversible reactions between ammonium, glutamate, and glutamine, and provide nitrogen necessary for the *de novo* synthesis of amino acids [11]. The carbon skeleton required to achieve this *de novo* synthesis can originate from the degradation of consumed amino acids or be provided through the CCM. Thus, carbon and nitrogen metabolisms are interconnected. The fluxes through these metabolisms are dependent of the amount of available nitrogen that affects the proportion of amino acids directly incorporated into the biomass, CCM contribution, and, finally, aroma production [14].

Currently, only little information is available about these non-*Saccharomyces* yeasts compared to *S. cerevisiae*. Their metabolism—especially nitrogen metabolism—has been addressed, determining the preferred and non-preferred nitrogen sources of several strains and pointing out specific features compared to *S. cerevisiae* [15,16,17]. In addition, underlying mechanisms have been investigated, either at the level of gene expression regulation in *Hanseniaspora vinae* [18] or in terms of fluxes partitioning in the metabolic network, explaining some specific metabolic traits of *Kluyveromyces marxianus* [19]. However, these works each focused on a single species, and cannot be extrapolated to other yeasts, due to the important variability between non-*Saccharomyces* strains regarding nitrogen resource management [20]. A better understanding of nitrogen metabolism in non-*Saccharomyces* species would help using them at their best potentiality during fermentation and optimise their co-culture with *S. cerevisiae*.

In this context, the aim of this study was to explore nitrogen resource management by the two non-*Saccharomyces* commercial yeasts most used in winemaking, viz. *Metschnikowia pulcherrima* Flavia and *Torulaspora delbrueckii* Biodiva, and thus gain information on the fate of the nitrogen sources. These strains have shown metabolic specificities compared to *S. cerevisiae*, especially for nitrogen preferences and volatile compounds production [8,17,21,22,23] that should be further explored. To this end, we applied a quantitative metabolic analysis approach using ^15^N- or ^13^C-labelled nitrogen sources to trace their partitioning through the metabolic network during alcoholic fermentation. Through the isotopic enrichment data, we identified the metabolic origin of both proteinogenic amino acids and volatile compounds, via the quantitative determination of nitrogen sources use. Our results provide essential information for a deeper understanding of nitrogen metabolism by non-*Saccharomyces* species and new insights for ultimately a better management of yeast nitrogen nutrition in co- or mixed-culture fermentation.

## 2. Materials and Methods

### 2.1. Yeast Strains and Media

The two non-*Saccharomyces* strains, *Torulaspora delbrueckii* (Biodiva, Lallemand) and *Metschnikowia pulcherrima* (Flavia, Lallemand), used in the study were propagated on yeast extract peptone dextrose (YPD) medium (2% glucose, 2% peptone, 1% yeast extract). Fermentations were carried out with a synthetic must (SM) that is similar to grape juice but with a defined composition, with modifications as in Bely et al. [24]. For the nitrogen source labelling experiment, the synthetic must contained 100 g/L glucose, 100 g/L fructose, and a mixture of 40% ammonium chloride and 60% amino acids as nitrogen sources (300 mg/L Yeast Assimilable Nitrogen, YAN). For the glucose labelling experiment, the SM contained 100 g/L glucose as the sole carbon source and ammonium chloride (300 mg/L YAN) as the nitrogen source. The concentration of organic acids, minerals, and vitamins in the SM were the same as those described by Su et al. [25]. SM was sterilised by filtration through 0.22 μm pore-size membrane filters (Labbox, Spain).

The labelled nitrogen sources, ^15^N ammonium chloride (99%), ^15^N_2_ L-glutamine (98%), U-^15^N_4_ L-arginine (98%), ^13^C_5_ L-valine (97–98%), ^13^C_6_ L-leucine (97–99%), and labelled ^13^C_6_ glucose (99%) were obtained from Euriso-top (Cambridge Isotope Laboratories, Saint-Aubin, France).

### 2.2. Fermentations and Sampling

A set of fermentations was carried out in 250 mL synthetic medium with the same defined chemical composition but with one nitrogen compound labelled at a time, among ^15^N-glutamine, ^15^N-arginine, ^15^N-NH_4_, ^13^C-valine, and ^13^C-leucine. For each labelling condition, duplicate cultures were performed, resulting in a total of 10 fermentations by strain. Samples were taken at four different stages (1/4, 1/2, 3/4, and maximum cell population, referred to as N_1/4_, N_1/2_, N_3/4_, and EN). Since *Torulaspora delbrueckii* and *Metschnikowia pulcherrima* have different fermentation rates, we applied different sampling points, i.e., 16 h, 20 h, 24 h, and 40 h for *T. delbrueckii*, and 12 h, 18 h, 24 h, and 40 h for *M. pulcherrima*.

### 2.3. Quantification of Consumed and Proteinogenic Amino Acids

Biomass weight was determined by filtering 10 mL well-mixed fermentation cultures through a pre-weighed 0.45 µm nitrocellulose filter. Filters were washed twice with 50 mL deionised water and dried at 105 °C for 48 h before weighing. Biomass was measured in duplicate for both yeast species.

The protein concentration of the biomass was quantified in duplicate using a bicinchoninic acid (BCA) protein assay kit (Sigma-Aldrich). Proteins were extracted from 1–2 mg biomass by incubation with 50% (v/v) dimethyl sulfoxide (DMSO) for 1 h at 105 °C and their concentration was determined following the manufacturer’s instructions.

In order to determine the amino acid composition in protein, we incubated cell pellets overnight in 10% (v/v) trichloracetic acid at −20 °C for protein precipitation. Proteins were then hydrolysed in 6 N HCl at 105 °C for 24 h. Amino acid concentrations were determined with a specific amino acid analyser (Biochrom 30, Biochrom) combining ion-exchange chromatography and spectrophotometric detection after ninhydrin revelation. The percentage of each amino acid in proteins was calculated by dividing the concentration of each amino acid by the total amount of amino acids in the protein extract.

The residual nitrogen compounds in the SM were analysed by Ultra High Performance Liquid Chromatography UHPLC equipped with a UV detector (Thermo Scientific, MA, USA). Samples were derivatised with diethylethoxymethylenemalonate (DEEMM). An accucore C18 (Thermo scientific) LC column was used to separate amino acids, with acetonitrile and acetate buffer as mobile phases, with the working gradient as described in Su et al. [25] (Table 1). Since 10 fermentations were carried out for each strain, we obtained the residual nitrogen data in 10 replicates.

### 2.4. Isotopic Enrichment of Intracellular Amino Acid Analysis

The biomass was hydrolysed by adding 1.2 mL 6 M HCl and incubating the samples for 16 h at 105 °C in tightly closed glass tubes in a dry heat oven. After incubation, we added 800 µL distilled water and the samples were centrifuged to remove cell debris. The supernatant was separated in four aliquots of 400 µL and were further dried at 105 °C until they reached a syrup-like state (4–5 h). These fractions were then utilised for amino acid derivatisation that was performed as described previously [13,14,19,26]. Two different derivatisation agents were used: (a) ethylchloroformate (ECF) derivatisation was carried out by first dissolving the syrup-like hydrolysate in 200 μL of 20 mM HCl and 133 μL of a pyridine-ethanol mixture (1:4), then adding 50 μL ECF. Derivatives were extracted with 500 μL dichloromethane, and centrifuged at 4 °C at 10,000× *g* for 5 min. The organic phase was carefully transferred into a 2 mL vial for GC–MS analysis. (b) *N*,*O*-Bis(trimethylsilyl)trifluoroacetamide (BSTFA) derivatisation was performed by adding 200 μL acetonitrile to the hydrolysate, and 200 μL BSTFA was then added to the dissolved hydrolysates to derivatise amino acids. The mixture was incubated 4 h at 135 °C and the organic phase was carefully transferred to a 2 mL vial for GC–MS analysis. The experiment was carried out in duplicate.

Derivatised samples were analysed using a Hewlett Packard 6890 gas chromatograph (Agilent Technologies, Santa Clara, CA, USA) equipped with a CTC Combi PAL Autosampler AOC-5000 (Shimadzu, Columbia, SC, USA) and coupled to an HP 5973 mass spectrometer. The gas chromatograph was fitted with a 30 m × 0.25 mm DB-17 ms column with a 0.15 μm film thickness (Agilent Technologies). Different GC–MS programs were used for the analysis of samples derivatised by ECF and BSTFA as described by Crépin et al. [13] and Bloem et al. [26]. The mass spectrometer quadrupole temperature was set at 150 °C. The source was set at 250 °C and 300 °C for BSTFA and ECF derivatives, respectively, and the transfer line held at 250 °C. The MS was operated in selected ion monitoring (SIM) mode with positive ion electron impact at 70 eV using the characteristic ions of amino acid fragments reported previously [13,26]. For each amino acid fragment, the outcome of the analysis was a cluster of intensities corresponding to its different mass isotopomers. These data were subsequently processed using IsoCor software developed by Millard et al. [27]; they were corrected for natural labelling and assessed for the isotopic enrichment of each amino acid, defined as the labelled fraction of a given compound (expressed as a percentage).

### 2.5. Isotopic Enrichment of Volatile Compounds Analysis

The experiment for isotopic enrichment of volatile compounds analysis was only carried out with fermentations containing ^13^C-leucine or ^13^C-valine. The volatile compounds were extracted with dichloromethane from 5 mL samples with deuterated internal standards. Extracted molecules were separated by a HP 6890 gas chromatograph (Agilent Technologies) equipped with a 30 m × 0.25 mm Phenomenex ZB-WAX-fused silica capillary column with a 0.25 µm film thickness (Agilent Technologies) and helium as the carrier gas using the procedure previously described [28]. Compounds were detected using an HP 5973 mass spectrometer in SIM mode with positive ion electron impact at 70 eV. The ion clusters reported were used for the quantification and the determination of the labelling patterns of volatile compounds in a previous study [14]. These ion clusters were selected on the basis of their high signal-to-noise ratio and low interference from other compounds. The concentration of each volatile molecule was quantified from the sum of the intensities of the corresponding ion cluster (Appendix A). In parallel, for each ion cluster, intensities were corrected for natural labelling using IsoCor software [27] and processed to assess to the isotopic enrichment of volatile compounds, defined as the fraction of labelled molecule with respect to its total production (expressed as a percentage). The isotopic enrichment of volatile compounds was determined in duplicate and the concentration of volatile compounds was determined in four replicates.

### 2.6. Outline of Experiment Design and Data Analysis

The experiment design, previously experienced and validated [13,14], relied on a set of 10 fermentations for each strains, using a synthetic medium with the same chemical composition. Fermentations were carried out labelling specifically one of the nitrogen sources of interest, namely, ^15^N-glutamine, ^15^N-arginine, ^15^N-NH_4_, ^13^C-valine, and ^13^C-leucine, each condition being carried out in duplicate. Consumed nitrogen amount was calculated by subtracting the residual nitrogen concentration at each sampling time from the initial nitrogen concentration. Biomass weight, protein concentration, and amino acid composition for each strain were determined from fermentations carried out using the same conditions but without labelled compound, because of the cost of these molecules [13], and allowed to calculate the concentration in proteinogenic amino acids. The labelled fraction of a protinogenic amino acid or a volatile compound was calculated by multiplying its concentration in millimolars by its isotopic enrichment. The difference from the total amount of the compound corresponded to the unlabelled part. Fluxes in the metabolic reactions involved in the synthesis of a target compound (proteinogenic amino acid or volatile molecule) from a labelled nitrogen source were quantified by dividing the labelled fraction of the compound by the total amount of consumed labelled molecule in millimolars. Calculations were done from mean values provided together with raw data and further details on the calculations in Appendix A. 

## 3. Results

To elucidate how *M. pulcherrima* and *T. delbrueckii* manage nitrogen resources during fermentation for growth but also for volatile compounds production, we assessed flux distribution in the nitrogen metabolic network using a quantitative approach relying on mass balance and isotopic tracer experiments. For each strain, we performed fermentations under the same conditions (medium, fermentation management), but with a different labelled nitrogen source each time. This design takes advantage of the high reproducibility of fermentations achieved under controlled conditions, and its relevance was demonstrated with *S. cerevisiae* [13,14].

### 3.1. Incorporation of Nitrogen from Glutamine, Ammonium, and Arginine into Proteinogenic Amino Acids

#### 3.1.1. Consumption of Glutamine, Ammonium, and Arginine during the Growth Phase

NH_4_Cl, arginine, and glutamine are the three major nitrogen sources in a synthetic medium, and more generally in grape juice. The ^15^N-labelled form of each compound helped us to explore the way in which the two non-*Saccharomyces* species redistribute nitrogen from these three main sources to fulfil their anabolic requirements.

Arginine, glutamine, and ammonium represented more than half of the nitrogen amount consumed by both strains during fermentation, with *T. delbrueckii* consuming nitrogen sources more efficiently than *M. pulcherrima* (Figure 1A and Figure 2A). When maximum population was reached, NH_4_Cl, arginine, and glutamine had been completely depleted by *T. delbrueckii*, while *M. pulcherrima* had only consumed 26%, 66%, and 79% of the initial NH_4_Cl, arginine, and glutamine contents, respectively. Even if the assimilation of all three sources started at the beginning of fermentation, their relative contribution to the total consumed nitrogen varied throughout the fermentation on the basis of the species. On one hand, *T. delbrueckii* assimilated glutamine, arginine, and ammonium in the same proportion during fermentation, similar to the proportion of these nitrogen sources in the medium. Consequently, ammonium was the largest nitrogen provider for *T. delbrueckii*, accounting for 29% of total consumed nitrogen, followed by glutamine (23%) and arginine (22%). On the other hand, *M. pulcherrima* showed faster consumption of glutamine and arginine than ammonium at the early stages of growth, although the contribution of ammonium to the intracellular pool gradually increased during growth. At the end of the growth phase, glutamine had provided 34% of total consumed nitrogen by *M. pulcherrima*, followed by arginine (27%), while ammonium was much less consumed than the other two sources, accounting for only 14% of total consumed nitrogen.

#### 3.1.2. Relationships between Consumption of Amino Acids and Anabolic Requirements in Proteinogenic Amino Acids

The first consideration was whether amino acids were always consumed in sufficient amounts to cover the needs of the corresponding proteinogenic amino acids. The ratio between consumed and the relative proteinogenic amino acids showed that, for *T. delbrueckii*, only a few amino acids (alanine, arginine, glutamine, methionine, and tryptophan) were consumed in sufficient amounts to cover the anabolic requirements throughout fermentation (ratio > 1 in Figure 1B). The consumption of the other amino acids was too low to cover the needs for protein biosynthesis, emphasising the importance of *de novo* amino acid synthesis and nitrogen redistribution through metabolism. The *M. pulcherrima* strain displayed a similar behaviour, but in this case, additional amino acids (viz. glutamate, leucine, isoleucine, histidine) were consumed in adequate amounts to fulfil anabolic requirements (Figure 2B). This difference can be explained by the limited growth of *M. pulcherrima*, which consequently resulted in a lower anabolic requirement for amino acids. The ratio obtained for valine with *M. pulcherrima* was prominent compared to the other amino acids, as this ratio increased from the first stages of the growth phase to be close to 1 at its end. This evolution could be the sign of a decorrelation between the consumption of this amino acid and the anabolic requirement of *M. pulcherrima*.

#### 3.1.3. Redistribution of Glutamine, Ammonium, and Arginine for De Novo Synthesis of Other Amino Acids

Being the most abundant nitrogen sources in the synthetic must, consumed NH_4_Cl, arginine, and glutamine were catabolised for the synthesis of other amino acids. By combining isotopic enrichment data from the three ^15^N-labelling experiments, we obtained a comprehensive picture of the main nitrogen distribution into proteinogenic amino acids (except for tryptophan, methionine, cysteine, and tyrosine, which are not detected by the GC–MS method used), and about the contribution of *de novo* synthesis to proteinogenic amino acids against direct incorporation (Figure 1C and Figure 2C). Irrespective of the labelled nitrogen source, ^15^N isotopic labelling was recovered in all the proteinogenic amino acids measured, even if the amount of amino acids consumed was greater than anabolic requirements. This was true for isoleucine, of which more than 56% was synthesised *de novo* by *M. pulcherrima*, while this species assimilated a higher amount than its needs for protein synthesis (Figure 2B,C). Interestingly, the ratio between the amount of nitrogen originating from arginine, glutamine, and ammonium was similar in all proteinogenic amino acids, with few exceptions only (Appendix A). First, assessing the contribution of glutamine and arginine compared to ammonium (Arg */NH4 *, Gln */Arg *, Figure 3), we observed that for *T. delbrueckii*, ammonium was the major contributor to the *de novo* synthesis of other amino acids (ratio < 1), followed by glutamine and then arginine. Nevertheless, for *M. pulcherrima*, glutamine was the highest contributor to amino acid *de novo* synthesis, especially during the early growth phase (ratio > 1). Ammonium was catabolised in greater amounts than arginine for the synthesis of other amino acids, although, overall, a higher amount of arginine was consumed. Furthermore, no correlation was found comparing the ratio between the consumption of amino acids and its proteinogenic content on one hand, and their enrichment pattern, on the other hand (Figure 1B, Figure 2B, and Figure 3B). From all these observations, taking into account the contribution of amino acid nitrogen to *de novo* synthesis, it appeared that respectively 78% and 80% of *M. pulcherrima* and *T. delbrueckii* proteinogenic amino acids were newly synthesised from a common nitrogen pool. 

#### 3.1.4. Different Isotopic Enrichment Patterns of Proteinogenic Amino Acids

The isotopic enrichment for each proteinogenic amino acid changed during all four fermentation stages, with patterns depending on the proteinogenic amino acid, strain, and labelled nitrogen source.

Aromatic, aliphatic, and hydroxyl amino acids displayed the same general profile of labelling incorporation throughout fermentation. When glutamine or ammonium were used as the labelled nitrogen source, isotopic enrichment in these proteinogenic amino acids in *T. delbrueckii* ranged between 14% and 20%, and 22% and 30%, respectively (Figure 3B). A slight decrease in labelling in the course of fermentation (less than 5%) was found for all these compounds. In the presence of labelled arginine, the initial isotopic enrichment in these classes of proteinogenic amino acids was lower, between 6% and 10%, but increased more substantially, by at least 10%, throughout the process. In the case of *M. pulcherrima*, the initial isotopic enrichment in aromatic, aliphatic, and hydroxyl proteinogenic amino acids was higher with ^15^N-glutamine (from 17% to 25%) compared to ^15^N-NH_4_Cl (from 4% to 10%) or ^15^N-arginine (between 1% and 7%). Furthermore, for this species, the contribution of nitrogen from glutamine to *de novo* synthesis of proteinogenic compounds decreased during fermentation, while those of ammonium and arginine largely increased, achieving an average of 22% and 12%, respectively, at the end of culture.

In general, the highest isotopic enrichment was measured in glutamate and aspartate, with a proportion of *de novo* synthesis from ammonium, glutamine, and arginine accounting for approximately 70% and 75% of the total amount of these proteinogenic amino acids for *M. pulcherrima* and *T. delbrueckii*, respectively (Figure 1C and Figure 2C). The contribution of glutamine to proteinogenic aspartate (31%) and glutamate (33%) was predominant in *M. pulcherrima*, while ammonium was the main nitrogen provider for *de novo* synthesis of aspartate (22%) and glutamate (22%) in *T. delbrueckii*.

Other exceptions to the general pattern concerned histidine and lysine. The first specificity to be highlighted for both NS strains was their significant anabolic requirements for lysine, with this amino acid accounting for a large fraction of proteins (10% of the total weight of proteinogenic amino acids in these species compared with less than 2% in *S. cerevisiae*; Figure 4 and Figure 5). As a consequence, even if the nitrogen fraction redistributed towards *de novo* lysine synthesis was relatively high in comparison with the other proteinogenic amino acids, we detected a low incorporation of nitrogen from arginine, glutamine, and ammonium in proteinogenic lysine (below 37% versus over 53% for the other amino acids) (Figure 1C and Figure 2C). Furthermore, the combination of *de novo* synthesis of proteinogenic lysine with the direct incorporation of consumed lysine (2.3 mg/L) was not sufficient to cover the anabolic requirements of either *T. delbrueckii* or *M. pulcherrima*. We also observed differences in the labelling pattern of proteinogenic histidine, revealing differences in the metabolic origin of proteinogenic histidine between *T. delbrueckii* and *M. pulcherrima* (Figure 4). Indeed, at least 52% of proteinogenic histidine was *de novo* synthesised using nitrogen from ammonium, arginine, and glutamine in *T. delbrueckii*, as opposed to a much smaller fraction (20%) in *M. pulcherrima*. Differences might correlate with the anabolic requirements for histidine, around four times higher in *T. delbrueckii* (234 µM) than in *M. pulcherrima* (83 µM), along with a delayed histidine consumption by *T. delbrueckii*. Thus, direct incorporation of histidine (83 µM) covered anabolic requirements in *M. pulcherrima* but not in *T. delbrueckii*.

#### 3.1.5. Metabolism of Arginine and Proline during Fermentation

The repartition of consumed arginine in the metabolic network was quite different from that of ammonium and glutamine that were efficiently recovered in proteins in the early stages of growth. First, the enrichment percentage of proteinogenic arginine was high (94% for *M. pulcherrima* and 87% for *T. delbrueckii*) when the maximum population was reached on a labelled arginine medium (Figure 4). This observation demonstrated that the main metabolic origin of proteinogenic arginine is a direct incorporation of consumed arginine, with a low contribution of *de novo* synthesis. However, at the end of the growth phase, we did not recover a large part of nitrogen from consumed arginine in proteins, but as arginine (direct incorporation: 30% and 17% of consumed arginine for *T. delbrueckii* and *M. pulcherrima* respectively) and other amino acids (redistribution of nitrogen through arginine catabolism and *de novo* synthesis: 30% of consumed arginine for *T. delbrueckii* and 16% for *M. pulcherrima*). Thus, the remaining arginine may have been intracellularly stored as nitrogen stock.

The incorporation pattern of labelled nitrogen in proteinogenic proline was also different from that of the others amino acids (Figure 1C and Figure 2C). In total, only 25% and 36% of proteinogenic proline nitrogen originated from the three main sources for *T. delbrueckii* and *M. pulcherrima*, respectively. This observation showed that *de novo* proline synthesis remained limited for both strains, suggesting a substantial direct incorporation of consumed proline into proteins. Interestingly, arginine emerged as the major nitrogen provider for proteinogenic proline synthesis (Figure 3A), at the expense of the two other sources, reflecting the formation of proline as an intermediate in arginine catabolism that is not further catabolised to contribute to the intracellular nitrogen pool. Finally, the labelling pattern of proteinogenic proline of *M. pulcherrima* and *T. delbrueckii* reflected the profile of arginine consumption by the two NS strains, with an early arginine uptake by *T. delbrueckii* (providing 25% of nitrogen for proline during the early growth phase, but only 13% at the end of growth phase) and, on the opposite, a delayed arginine assimilation by *M. pulcherrima* resulting in a progressive increase in the proportion of nitrogen from labelled arginine in proteinogenic proline (Figure 3B).

### 3.2. Origin of the Carbon Backbone of Proteinogenic Aliphatic Amino Acids

Amino acid catabolism not only contributes to the internal nitrogen pool, but also provides the carbon backbone for cellular biosynthesis as well as precursors for aroma production. The fate of the carbon backbone of the branched chain amino acids valine and leucine was explored using ^13^C-labelled amino acids. These nitrogen sources were totally consumed by both yeast strains during wine fermentation, with an earlier onstart of leucine consumption.

#### 3.2.1. Consumed Aliphatic Amino Acids Recovered in Proteinogenic Amino Acids

The amount of labelled carbon in proteinogenic leucine and valine was measured to determine the proportion of these molecules originating from the direct incorporation of their respective consumed amino acids (Figure 6 and Figure 7). A low isotopic enrichment of proteinogenic valine was observed, indicating a limited direct incorporation of valine into proteins (below 25% and 14% for *M. pulcherrima* and *T. delbrueckii*, respectively) and an important contribution of *de novo* synthesis via precursors from CCM. Incorporation profiles differed between the two studied strains: the enrichment percentage stayed constant at approximately 14% during fermentation for *T. delbrueckii*. For *M. pulcherrima*, by contrast, the enrichment percentage was higher in the early growth phase, between 31% and 38%, before decreasing to 25% when the population reached its maximum (Appendix A).

A large fraction of consumed leucine was incorporated into proteinogenic leucine in both yeasts. Throughout fermentation, the part of consumed leucine directed toward its corresponding proteinogenic amino acid in *T. delbrueckii* and *M. pulcherrima* ranged from 84% to 64%, and from 84% to 54%, respectively (Appendix A). As a consequence, isotopic enrichment in proteinogenic leucine was substantially higher than for valine, with 40% and 21% enrichment for *M. pulcherrima* and *T. delbrueckii*, respectively (Figure 6 and Figure 7). However, CCM contribution was still the main source of carbon for proteinogenic leucine synthesis, except during *M. pulcherrima* early growth phase in which the direct incorporation of consumed leucine covered up to 78% of leucine anabolic requirements. The absolute amount of consumed leucine recovered in proteins was identical in *M. pulcherrima* and *T. delbrueckii* fermentations. However, the contribution of consumed leucine to the total leucine proteinogenic pool was lower in *T. delbrueckii*, due to higher anabolic requirements, which resulted in a larger *de novo* synthesis of leucine from unlabelled precursors of the CCM.

#### 3.2.2. Formation of Volatile Compounds through Valine and Leucine

To draw up a complete picture of the fate of consumed valine and leucine within cells, we measured isotopic enrichment in the volatile compounds deriving from the assimilation of these molecules through the Ehrlich pathway, in addition to that of their proteinogenic counterparts (Figure 6 and Figure 7, and Appendix A). Valine catabolism through the Ehrlich pathway accounted for a large part of the fate of this amino acids for the two studied species. For *M. pulcherrima*, more than 38% of consumed valine was directed towards the production of volatile compounds deriving from α-ketoisovalerate (isobutanol, isobutyric acid). As a consequence, the fraction of isobutanol produced from valine catabolism (assessed from the isotopic enrichment) reached up to 40% (N_1/2_) when most labelled valine was consumed. The further decrease in isotopic enrichment of isobutanol to 16% combined with the increase in total isobutanol production revealed that this higher alcohol was synthesised from CCM precursors during the last growth stages. The conversion between α-ketoisovalerate and α-ketoisocaproate was relatively low, which led to a low overall contribution of valine to the production of isoamyl alcohol (isotopic enrichment 3%) and proteinogenic leucine (isotopic enrichment 5.4%) (Appendix A).

Flux distribution through the metabolic network from consumed valine in *T. delbrueckii* was, however, different. First, in this species, most α-ketoisovalerate produced from the catabolism of consumed valine was directed toward the production of α-ketoisocaproate (18%) at the expense of volatile compounds formation, with isobutanol and isobutyric acid respectively accounting for only 7% and 1% of consumed valine. Furthermore, a large fraction of α-ketoisocaproate from valine assimilation, accounting for 16.2% at the end of the growth phase, was directed towards the production of proteinogenic leucine, while the formation of isoamyl alcohol was limited (3% of consumed valine).

Since leucine was mostly used directly as protein building blocks, only a limited proportion of consumed leucine was catabolised. Interestingly, the proportion of consumed leucine directed towards the production of isoamyl alcohol was higher in *M. pulcherrima* (6% of consumed leucine) than in *T. delbrueckii* (2%). Consequently, isotopic enrichment in isoamyl alcohol during *M. pulcherrima*, of over 11%, was particularly high compared to those measured with *T. delbrueckii* or previously reported with *S. cerevisiae* fermentations [13]. A substantial fraction of isoamyl alcohol produced by *M. pulcherrima* originated from consumed leucine, indicating that leucine availability in the medium may directly affect the formation of isoamyl alcohol during the early growth phase.

The analysis of the compounds presented here allowed to recover 66% of leucine consumed by *T. delbrueckii* and only 59% of consumed leucine in *M. pulcherrima* at the end of the growth phase. This suggests that a fraction of leucine may be stored or converted into others compounds.

## 4. Discussion

Yeast nitrogen metabolism is of significant importance for wine fermentation. Indeed, it influences both fermentation rate and duration and participates in the formation of volatile compounds which in turn determine wine quality. Although there is an increasingly frequent use of non-*Saccharomyces* strains for winemaking, little information is available regarding nitrogen metabolism in these species, because they have long been considered spoilage organisms only and thus not as studied as *S. cerevisiae*. Here, the exploration of the metabolic network of two non-*Saccharomyces* species particularly relevant to winemaking, *T. delbrueckii* and *M. pulcherrima,* by tracing ^15^N- or ^13^C-labelled nitrogen sources partitioning revealed their specificities in the management of nitrogen resource during fermentation.

First, the inability of *M. pulcherrima* to consume 300 mg/L YAN during fermentation (50% consumption only) must be balanced against the capacity of *T. delbrueckii* to deplete all available nitrogen. This has to be considered with respect to the lower anabolic requirement of *M. pulcherrima* (final biomass content: 1.6 g/L) compared to *T. delbrueckii* (4.1 g/L). This lower biomass production by *M. pulcherrima* during fermentation was also observed by Roca-Mesa et al. [29]. Furthermore, while *T*. *delbrueckii* consumed all three major nitrogen sources (arginine, glutamine, ammonium) with almost the same efficiency [17,30], *M*. *pulcherrima* first imported glutamine, then arginine, at the expense of ammonium. As a result, only 26% of initial NH_4_Cl had been consumed by *M*. *pulcherrima* at the end of the growth phase. A late and incomplete consumption of NH_4_Cl has also been reported in another non-*Saccharomyces* yeast, *Kluyveromyces marxianus* [19], that is connected with the genetic background of this species. While three ammonium transporters have been identified in *S. cerevisiae* with high (Mep2p, Mep1p) or low (Mep3p) affinity [31,32], only one gene coding for an ammonium transporter, orthologous to the *S. cerevisiae MEP3* gene, has been described in *K. marxianus*. Regarding *M. pulcherrima*, automatic annotation approaches have only identified a candidate gene for ammonium transport, orthologous to *MEP2* [33], and further investigations are required to elucidate why this species displays a poor capacity to uptake this nitrogen source (number of transporters, intrinsic properties, regulation of gene expression). Regarding the use of *non-Saccharomyces* yeasts with *S. cerevisiae* in sequential fermentation, it is clear that the low nitrogen consumer *M. pulcherrima* has a limited impact on *S. cerevisiae* growth, while *T. pulcherrima* may deplete some nitrogen sources in the medium, preventing the implantation of *S. cerevisiae,* as previously reported [8,13].

As previously observed with *S. cerevisiae* [13], the direct incorporation of consumed amino acids into proteins is not at a high enough amount to fulfil anabolic requirements in *T. delbrueckii* or *M. pulcherrima*. Therefore, *de novo* synthesis of proteinogenic amino acids takes place in the non-*Saccharomyces* species, involving a redistribution of nitrogen from the consumed nitrogen sources—in particular from the three most abundant ones—towards the proteinogenic amino acids. Thus, ammonium- and glutamine-derived nitrogen was recovered in all proteinogenic amino acids, at an average level of 18% and 21% (for ammonium and glutamine, respectively) of the total concentration of proteinogenic amino acids in *M. pulcherrima* and 25% and 15%, respectively, in *T. delbrueckii*. It is important to point out that *T. delbrueckii* mostly used ammonium-derived nitrogen for the *de novo* synthesis of amino acids preferentially to glutamine, contrary to *S. cerevisiae*, which uses both sources equally (22% and 20%, respectively), suggesting a different management of the nitrogen central core between the two species. Conversely, less than 18% of nitrogen in *M. pulcherrima* proteinogenic amino acids originated from consumed ammonium, in line with the low capacity of this species to take up this compound.

Regarding the third major nitrogen source, we found that most proteinogenic arginine originated from consumed arginine for both strains, as previously reported for *S. cerevisiae*. However, the isotopic enrichment of proteinogenic arginine in *M. pulcherrima* and *T. delbrueckii*, varying from 87% to 94% depending on fermentation stage, was slightly lower than that previously reported for *S. cerevisiae*, which remained constant at 98% throughout fermentation [13]. Furthermore, labelled nitrogen was found in proteinogenic arginine during fermentations carried out in the presence of either ^15^N-glutamine or ^15^N-ammonium, consistent with a low but effective *de novo* synthesis of arginine (up to 13%) in *M. pulcherrima*. In *S. cerevisiae*, the ArgR/Mcm1 pathway, composed of the *ARG1,3–6,8* genes and responsible for the synthesis of ornithine, citrulline, and arginine from glutamine and glutamate, is strongly repressed by cytoplasmic arginine [10,34]. Our observations likely indicate a different regulation of this metabolic route, not totally repressed by intracellular arginine concentration in *M. pulcherrima* and *T. delbrueckii,* as previously reported for *K. marxianus* [19]. Surprisingly, a part of consumed arginine was not recovered in proteins, accounting for 23% and 72% of arginine consumed by *T. delbrueckii* and *M. pulcherrima,* respectively. The most likely explanation for this discrepancy is a vacuolar storage of part of consumed arginine, combined with its possible further use to support growth after depletion of the other nitrogen sources [35,36]. Correspondingly, because of its limited growth, the anabolic requirements of *M. pulcherrima* are lower than those of *S. cerevisiae* or *T. delbrueckii.* Consequently, arginine catabolism (quite costly in ATP, cofactors, and NADPH, see [35,37]), to provide nitrogen for *de novo* synthesis, appears unnecessary as long as other nitrogen sources are intracellularly available. As regards *T. delbrueckii,* an increase in the proportion of labelled nitrogen from consumed arginine was observed in proteinogenic amino acids at the end of the growth phase, when most nitrogen compounds were depleted. This is in line with a remobilisation of previously stored arginine to fulfil nitrogen anabolic requirements and sustain further growth.

Overall, compared with the other amino acids, the part of *de novo* synthesised histidine and lysine was low, as shown by the weak isotopic enrichment of these compounds measured throughout *T. delbrueckii* and *M. pulcherrima* fermentations in the presence of labelled ammonium, glutamine, or arginine. This difference could be explained by a significant direct incorporation into proteins of their counterparts taken from the medium. Indeed, the inability of the two NS species to use histidine to support growth has been previously demonstrated [25]. This work also reported that *T. delbrueckii* displayed reduced fermentative activity when lysine was provided as the sole nitrogen source, while *M. pulcherrima* was simply unable to grow under these conditions. These observations suggested that *M. pulcherrima* and *T. delbrueckii* are unable to catabolise lysine or histidine, as already shown for *S. cerevisiae* [38,39]. Thus the direct incorporation into proteins of consumed lysine and histidine could be more beneficial to yeast than a *de novo* synthesis of these molecules, which is energy- and co-factor-consuming [40,41]. This is particularly relevant for proteinogenic histidine in *M. pulcherrima*, with over 75% direct incorporation of consumed histidine. However, this applies to a lesser extent in the case of the metabolic origin of histidine in *T. delbrueckii*, for which isotopic enrichment was higher, suggesting more efficient or alternative pathways for histidine synthesis. Finally, regarding the metabolic origin of proteinogenic lysine, overall amounts of consumed lysine combined with *de novo* synthesis using glutamine-, arginine-, and ammonium-derived nitrogen, which is less efficient compared to the other amino acids, were not sufficient to meet anabolic requirements. This suggests specific alternative pathways for lysine synthesis in these yeasts.

Isotope tracer experiments using ^13^C-labelled valine and leucine allowed for the elucidation of their fate, at least in part, in *M. pulcherrima* and *T. delbrueckii*. A general pattern similar to that of *S. cerevisiae* [13] was shown, with a substantial contribution of catabolism accounting for at least 36% of the amino acids consumed. However, the partitioning and fate of consumed leucine and valine in the metabolic network differed between the two strains and with *S. cerevisiae*, reflecting a different management of these nitrogen resources.

First, high yields of isobutanol and isoamyl alcohol were found during *M. pulcherrima* fermentations, combined with a higher isotopic enrichment in these volatile molecules compared with those measured with *S. cerevisiae* and *T. delbrueckii*. These observations showed that, in this strain, consumed amino acids were mostly catabolised. This behaviour is consistent with the limited growth of *M*. *pulcherrima* during wine fermentation, and consequently, with its low anabolic requirements in terms of amino acids. The first consequence was the decrease in the formation of ketoacids from CCM as precursors of *de novo* synthesis of amino acids, as shown by the higher isotopic enrichment of volatile molecules produced by this strain (10%) compared to that measured during *S. cerevisiae* or *T. delbrueckii* wine fermentations (1%). Then, the formation of volatile molecules through the Ehrlich pathway is likely an efficient way to eliminate ketoacids, toxic compounds for yeasts [42]. The substantial contribution of leucine and valine catabolism to the production of volatile compounds by *M. pulcherrima* is of technological interest. Therefore, it seems possible to modulate these compounds through nitrogen nutrition management in *M. pulcherrima* when it is used with *S. cerevisiae* (in mixed or sequential inoculation). Finally, the low isotopic enrichment measured in proteinogenic leucine and isoamyl alcohol during *M*. *pulcherrima* fermentation in presence of labelled valine revealed a limited capacity of this strain to convert α-ketoisovalerate to α-ketoisocaproate. It may be explained by a deficit in this strain in acetyl-CoA, which is used as co-substrate in the first step of the conversion of α-ketoisovalerate into α-ketoisocaproate by *LEU4* and *LEU9* [43]. We had already assumed that the intracellular availability of acetyl-CoA was limited in this strain, because of its low ability to produce acetate, acetate esters, and medium-chain fatty acids and their ethyl ester derivatives [44].

The behaviour of *T. delbrueckii* was quite different and in agreement with its efficient and important growth capacities. Accordingly, this species displayed low production of volatile compounds, and the labelling provided by valine or leucine was mainly recovered in their proteinogenic counterparts, at the expense of higher alcohols. This flux partitioning revealed that *T. delbrueckii* showed an optimal management of nitrogen resources to fulfil anabolic requirements, mainly directing consumed nitrogen sources toward protein formation.

Mass and isotope balance calculations revealed that only 72% and 64% of the labelling originating from consumed leucine and valine, respectively, during *T. delbrueckii* fermentation were recovered (in proteinogenic amino acids, higher alcohols, acetate esters, and branched amino acids); similarly, small fractions (50% and 65%, respectively) were recovered from *M. pulcherrima* fermentations. To account for these losses, stripping and evaporation of a fraction of the analysed compounds have first to be considered [45]. Indeed, up to 0.7% isoamyl alcohol and 21% isoamyl acetate can be lost to evaporation at 24 °C, i.e., the fermenting temperature used in our experiments. In addition to these analytical limitations, other metabolic routes may be considered for valine and leucine assimilation, leading to the production of metabolites not measured in this work. Thus, a previous study showed that *M. pulcherrima* was able to produce high concentrations of propyl acetate and ethyl propionate when fermenting with leucine as the sole nitrogen source [8]. Other metabolic pathways described to be involved in valine and leucine degradation in other organisms may also be operative in non-*Saccharomyces* species—such an example is terpenoid backbone biosynthesis through the mevalonate pathway, connected to leucine degradation in archaea, eukaryote, and bacteria [46,47]. Valine and leucine degradation can also provide precursors for the biosynthesis of macrolides (molecules such as erythromycin with antibiotic activity) in bacteria and fungi [48,49]. Furthermore, one of the characteristic features of *M. pulcherrima* is its ability to produce pulcherrimin, a red iron-containing pigment with antifungal and antibacterial properties [50]. The formation of its precursor, pulcherriminic acid, involving the cyclisation of two molecules of leucine, may be one of the outcomes of consumed leucine in *M. pulcherrima* [51,52]. Finally, an intracellular storage of leucine and valine, without further degradation, as reported for arginine, cannot be completely ruled out.

## 5. Conclusions

Isotopic tracer experiments using ^15^N- and ^13^C-labelled nitrogen sources allowed us to comprehensively explore nitrogen metabolism in two non-*Saccharomyces* species, *T. delbrueckii* and *M. pulcherrima.* As previously reported for *S. cerevisiae*, consumed nitrogen sources were mainly catabolised for the *de novo* synthesis of amino acids, although arginine, and to a lesser extent histidine and lysine, were mainly incorporated directly into proteins. Differences between strains were observed, as *T. delbrueckii* used ammonium preferentially to glutamine, while ammonium incorporation was low in *M. pulcherrima.* This knowledge is of technological interest, in order to define appropriate nitrogen nutrition strategies during sequential or co-inoculation fermentation of these non-*Saccharomyces species* in combination with *S. cerevisiae.*

Furthermore, important differences between strains in the partitioning of nitrogen compounds in the metabolic network were revealed, reflecting distinct nitrogen resource management by each species adapted to their anabolic requirements. An efficient use of nitrogen from consumed nitrogen sources towards *de novo* synthesis of proteinogenic amino acids in lieu of volatile compounds formation was found in *T. delbrueckii*, in line with this species’ high growth capacity. Conversely, in the low biomass-producing species *M. pulcherrima*, amino acids were directed toward catabolism, producing high amounts of ketoacids subsequently transformed into higher alcohols. Overall, these observations underline flux distribution significance within the metabolic network to understand the phenotypic behaviour of these yeast species during fermentation. This information is essential for the design of strategies allowing to modulate wine flavour profiles through the use of these species in co- or sequential fermentation with *S. cerevisiae*; however, further investigations are needed to take into account the metabolic consequences of interactions between the non-*Saccharomyces* species and *S. cerevisiae* strains.

## Figures and Tables

**Figure 1 microorganisms-08-00904-f001:**
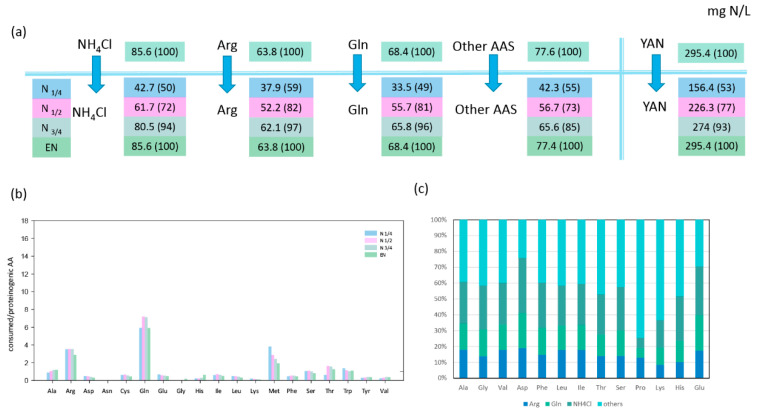
Nitrogen source consumption and redistribution in *Torulaspora delbrueckii*. (**a**) Nitrogen consumption (mg N/L) at four growth stages: N_1/4_, N_1/2_, N_3/4_, and EN. The values in brackets represent the percentage of consumed nitrogen sources to the initial concentration in the media. (**b**) Ratio between the concentration of consumed amino acids and the concentration of the corresponding proteinogenic amino acids (mM/mM) at different fermentation stages. Ratio > 1 indicate that the consumed amino acids are theoretically in sufficient quantities to meet anabolic requirements. (**c**) proportion (%) of labelled arginine, glutamine, and NH_4_Cl recovered into proteinogenic amino acid at the end of the growth phase (EN). Raw data and calculations are provided in Appendix A.

**Figure 2 microorganisms-08-00904-f002:**
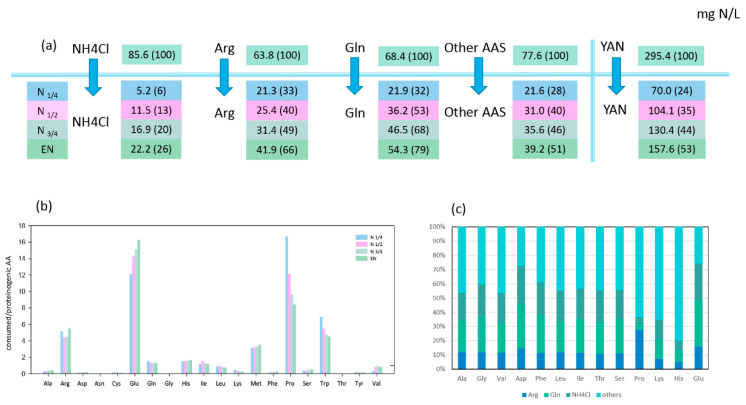
Nitrogen source consumption and redistribution in *Metschnikowia pulcherrima*. (**a**) Nitrogen consumption (mg N/L) during four growth stages: N1/4, N1/2, N3/4, and EN. The values in brackets represent the percentage of consumed nitrogen sources to the initial concentration in the media. (**b**) Ratio between the concentration of consumed amino acids and the concentration of the corresponding proteinogenic amino acids (mM/mM) at different stages of fermentation. Ratio > 1 indicate that the consumed amino acids are theoretically in sufficient quantities to meet anabolic requirements. (**c**) Proportion (%) of labelled arginine, glutamine, and NH_4_Cl recovered into proteinogenic amino acid at the end of the growth phase (EN). Raw data and calculations are provided in Appendix A.

**Figure 3 microorganisms-08-00904-f003:**
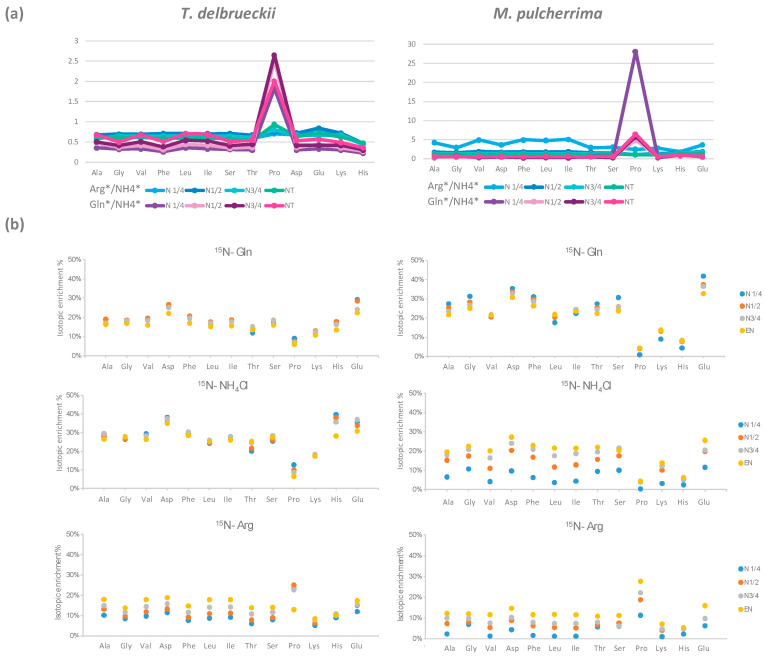
(**a**) Ratios between the amount of nitrogen from arginine and the amount of nitrogen from NH_4_Cl, Arg */NH_4_ *, and the amount of nitrogen from glutamine and the amount of nitrogen from NH_4_Cl, Gln */NH_4_ *, recovered in each proteinogenic amino acid. (**b**) Isotopic enrichment of proteinogenic amino acids at the four stages of growth when glutamine, NH_4_Cl, or arginine is ^15^N-labelled. Raw data used for the calculations are provided in Appendix A.

**Figure 4 microorganisms-08-00904-f004:**
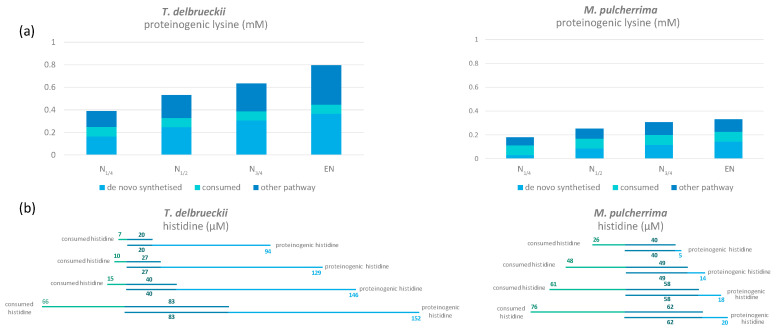
(**a**) Amount of proteinogenic lysine coming from *de novo* synthesis (light blue), directly incorporated from consumed lysine (green) and other pathways (dark blue) in *T. delbrueckii* and *M. pulcherrima*. (**b**) Fate of histidine in *T. delbrueckii* and *M. pulcherrima*. The amount of direct incorporation from consumed histidine to proteinogenic histidine is in dark blue; the rest of the consumed histidine is represented in green, while *de novo* synthesised histidine is in light blue. Raw data used for the calculations are provided in Appendix A.

**Figure 5 microorganisms-08-00904-f005:**
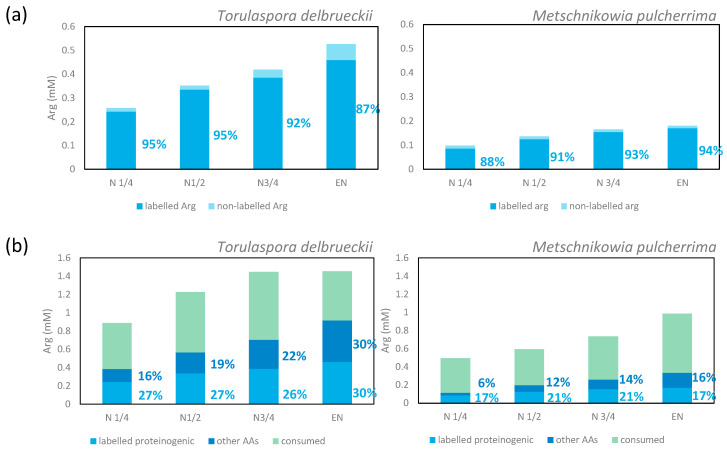
(**a**) Origin of proteinogenic arginine. The labels show the percentage of labelled arginine in total proteinogenic arginine (isotopic enrichment). (**b**) Fate of consumed arginine. The labelled arginine consumed was incorporated directly into proteinogenic arginine (light blue bar) or catabolised for the synthesis of other amino acids (dark blue bar). The labels indicate percentages in the total amount of consumed arginine. Data were calculated from the mean value. The detailed information is shown in Appendix A.

**Figure 6 microorganisms-08-00904-f006:**
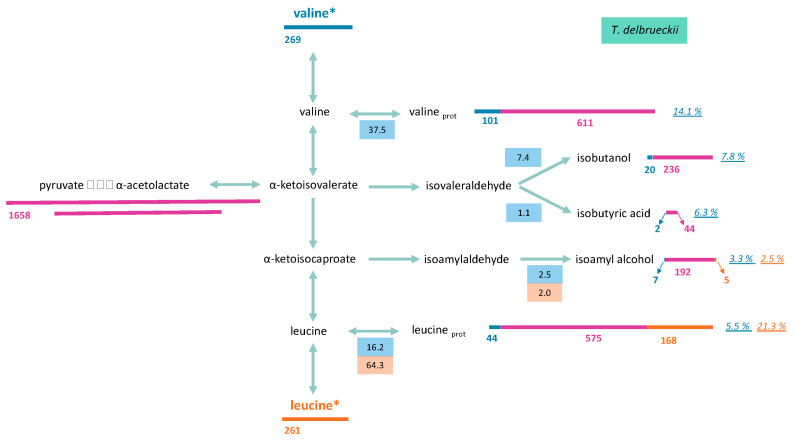
Metabolism of valine and leucine of *T. delbrueckii* at the end of growth phase. Labelled valine (valine*) or leucine (leucine*) were used during fermentations. Length of bars (blue: valine; orange: leucine; pink: central carbon metabolism) is proportional to the compound concentration (μM). Values in blue and orange boxes represent the fraction of consumed valine and leucine (µM) distributed through the pathway, respectively. The italic numbers indicate the percentage of each molecule (proteinogenic amino acid or volatile compound) originated from valine (blue) or leucine (orange). Raw data and details on calculation procedure are provided in Appendix A.

**Figure 7 microorganisms-08-00904-f007:**
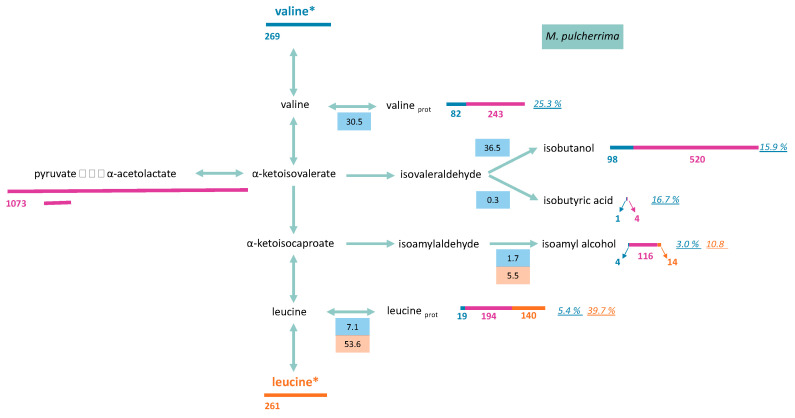
Metabolism of valine and leucine of *M. pulcherrima* at the end of growth phase (μM). Labelled valine (valine*) or leucine (leucine*) were used during fermentations. Length of bars (blue: valine; orange: leucine; pink: central carbon metabolism) is proportional to the compound concentration (μM). Values in blue and orange boxes represent the fraction of consumed valine and leucine (µM) distributed through the pathway, respectively. The italic numbers indicate the percentage of each molecule (proteinogenic amino acid or volatile compound) originated from valine (blue) or leucine (orange). Raw data and details on calculation procedure are provided in Appendix A.

**Table 1 microorganisms-08-00904-t001:** Eluent gradient for the HPLC determination of amino acids.

Time (min)	0.0	3.0	5.0	11.0	12.5	14.0	18.0	21.0	23.0	25.0	26.0
Phase A (%)	95.0	94.0	92.0	90.0	88.0	82.0	80.0	70.0	60.0	25.0	20.0
Phase B (%)	5.0	6.0	8.0	10.0	12.0	18.0	20.0	30.0	40.0	75.0	80.0

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
