# Peer review of "Isotopic Tracers Unveil Distinct Fates for Nitrogen Sources during Wine Fermentation with Two Non-Saccharomyces Strains"

_microorganisms, 2020, doi:10.3390/microorganisms8060904_

Round 1

Reviewer 1 Report

Interesting paper however the authors seem to have left out important results in supplementary files that no-one will read. These files need to come in.

Moreover, Figure 5 needs to be more clear regarding colours I would suggest adding arrows to where different colours are indicated. Colours are not really visible especially light green and light blue. So denoting with arrows and labelling them will solve the problem.

Figure 4 needs to have statistical analysis incorporated.

However, the authors seem to have missed design of the experiments. No statistical methodology is mentioned in the materials and methods section.

They just mention that 'The concentration of each volatile molecule was quantified from the sum of the intensities of the corresponding ion cluster'.

No statistical significance of the compounds is mentioned.

English needs to be corrected throughout the text

e.g. reflecting distinct nitrogen resource managements?

low capacity of this species to uptake this compound?

Calculating mass and isotopic balances revealed?

This suggests that a fraction of leucine may be stored or converted into others compounds. Which are these compounds? What other authors discuss?

I would agree that these experiments are early and that further investigation is required regarding the metabolic consequences of interactions between the non-Saccharomyces species and S. cerevisiae strains.

Author Response

  What do you want to do ? New mailCopy

Reviewer 2 Report

Comments to Author

The current submission describes the nitrogen metabolism pattern of T. delbrueckii and M. pulcherrima. Results show that T. delbrueckii with high growth capacity; M. pulcherrima produce high amounts of ketoacids subsequently transformed into higher alcohols. The experiment design are reasonable and results are sound. In my opinion, the ms can be accepted after minor revision.

  1. The different species of non-Saccharomyces show the different metabolic consequence. The competition for nutrients in non-Saccharomyces and cerevisiae should be describe in the discussion section.
  2. In this research, which strain is more suitable for winemaking?
  3. The authors should compare their results with other similar literature for the tests carried out in this study.
  4. The method of UHPLC working gradient should be described in the methodology.
  5. The format of units are not correct, such as mL.

Author Response

  What do you want to do ? New mailCopy

Round 2

Reviewer 1 Report

The authors have addreessed all raised comments and article can now be accepted for publication after minor spell check.

e.g. p. 384

The detailed information is showed (shown) in supplementary material 1 & 2